# Calcium Intake and Health

**DOI:** 10.3390/nu11071606

**Published:** 2019-07-15

**Authors:** Gabriela Cormick, Jose M Belizán

**Affiliations:** 1Department of Mother and Child Health Research, Institute for Clinical Effectiveness and Health Policy (IECS-CONICET), Emilio Ravignani 2024, Buenos Aires 1414, Argentina; 2Department of Human Biology, Faculty of Health Sciences, University of Cape Town, Cape Town 7725, South Africa; 3Departamento de Salud, Universidad Nacional de La Matanza, San Justo 1903, Argentina

**Keywords:** calcium intake, calcium, health, hypertensive disorders, fortification

## Abstract

There are striking inequities in calcium intake between rich and poor populations. Appropriate calcium intake has shown many health benefits, such as reduction of hypertensive disorders of pregnancy, lower blood pressure particularly among young people, prevention of osteoporosis and colorectal adenomas, lower cholesterol values, and lower blood pressure in the progeny of mothers taking sufficient calcium during pregnancy. Studies have refuted some calcium supplementation side effects like damage to the iron status, formation of renal stones and myocardial infarction in older people. Attention should be given to bone resorption in post-partum women after calcium supplementation withdrawal. Mechanisms linking low calcium intake and blood pressure are mediated by parathyroid hormone raise that increases intracellular calcium in vascular smooth muscle cells leading to vasoconstriction. At the population level, an increase of around 400–500 mg/day could reduce the differences in calcium intake between high- and middle-low-income countries. The fortification of food and water seems a possible strategy to reach this goal.

## 1. Introduction

Calcium is a mineral involved in a large number of vital functions [1,2]. Although research on the role of calcium has been primarily focused on bone health, the effects of either dietary calcium or calcium supplements have been oriented towards other health outcomes lately. An observation made in the 1980s drew attention to the relationship between calcium intake and preeclampsia/eclampsia during pregnancy [3]. This originated from the evaluation of the Mayan diet in Guatemala consisting of soaking and cooking corn with limewater before grinding and the consequent high intake of calcium found to be associated with a low frequency of preeclampsia/eclampsia [4]. The objective of this article is to update the various effects of calcium on health supported by findings in randomised controlled trials (RCTs) with considerations of its availability and intake and propose suggestions for strategies to achieve adequate intake.

## 2. Sources of Calcium

Calcium intake is usually associated with the intake of dairy products such as milk, yogurt and cheese, as they are rich sources of calcium. Calcium-rich foods are dairy products, especially hard cheese that can provide 1 g of calcium per 100 g, whereas milk and yogurt can provide between 100 mg to 180 mg per 100 g. Cereals usually have around 30 mg per 100 g, however if they are fortified, the amount can reach 180 mg per 100 g. Nuts and seeds are also rich in calcium, especially almonds, sesame and chia that can provide between 250 to 600 mg per 100 g. Vegetables rich in calcium are kale, broccoli and watercress, which provide between 100 and 150 mg per 100 g [5]. However, the impact that these foods have on total calcium intake depends on the population food consumption patterns. Whereas dairy products represent around 14% of total dietary energy intake in developed countries, they represent only around 4% of total energy intake in developing countries [6]. In this way, some Asian countries have higher proportion of total calcium intake from non-animal foods such as vegetables, legumes and grains than from dairy products, though they also have a much lower calcium intake [6]. In the United states and in Holland, 72 and 58% of calcium supply come from dairy products, respectively, whereas in China, only around 7% of total calcium intake comes from dairy products, while most comes from vegetables (30.2%) and legumes (16.7%) [7,8,9] Fortified foods such as cereals and juices can additionally become important sources of calcium.

Supplements are also a great dietary source of calcium for some populations. Some calcium supplements, available with no prescription, have up to 1000 mg of calcium per tablet, which represents the nutritional requirements for most adults. However, the use of supplements also varies between countries. In the United States and Canada, around 40% of the adult population was reported to have taken calcium supplements in the month before the interview, and this figure increased to 70% in the older women group [3,6,10]. On the other hand, in Argentina and in Holland, very few women reported taking calcium supplements, even during pregnancy [5,11,12].

## 3. Calcium Recommendations

Calcium requirements are high during all stages of life [13]. Dietary reference values for individuals over 19 years of age vary from 1000 mg to 1300 mg, depending on the reference guidelines [1,2,14,15,16] (Table 1).

The dietary reference values are established to account for the needs of growth, development, functioning and health maintenance [15]. The requirements are calculated on the basis of a selected health outcome. The USA Institute of Medicine (IOM) conducted a review to assess the effect of calcium on health so as to update the dietary reference intakes for calcium in 2010 [2]. The review finally recommended to base the requirements for calcium on its effects on bone health and concluded that the evidence of the effect of calcium on cancer, cardiovascular disease, diabetes and autoimmune disorders was inconsistent, inconclusive as to causality and insufficient to inform nutritional requirements [7]. Besides, in order to establish calcium intake upper limits, the review evaluated the following clinical outcomes: all-cause mortality, cancer (incidence and mortality), soft tissue calcification, renal outcomes and adverse events reported in RCTs. No evidence of association was found in this report for mortality, soft tissue or cancer. An increased risk of renal stone [Hazard Ratio 1.17 (95% CI 1.02–1.34)] was reported in only one trial in women aged 50 to 79 years who received vitamin D3 400 IU supplements in combination with 1000 mg of calcium supplements [17]. Of the 63 included RCTs, 10 reported adverse events, mainly related to higher gastrointestinal discomfort in the groups receiving calcium and or Vitamin D supplements. The upper limit was set to 2500 mg/day for people aged 19 to 50 years and to 2000 mg/day for older people.

During pregnancy, most guidelines acknowledge the increased demand of calcium; however, while some guidelines increase recommendations up to 1300 mg/day to achieve a positive balance, other guidelines state that metabolic adaptations during pregnancy compensate the required calcium demand [2,16,18,19,20]. The IOM set in 2010 the dietary intake upper limit for pregnant women to 3000 mg/day for those aged 14 to 18 years and to 2500 for older ones [7], whereas the FAO/WHO recommendation of 2001 acknowledged that the risk of kidney stones from dietary hypercalciuria might be negligible and set the upper limit to 3000 mg/day independent of age. Since 2013, WHO recommends that all pregnant women from areas of low dietary calcium intake receive calcium supplementation from 1500 to 2000 mg/day from 20 weeks´ gestation, as evidence from randomized control trials shows a reduction of the risk of preeclampsia [21,22].

## 4. Global Calcium Intake. Inequities

In most low- and middle-income countries, the daily calcium intake is well below recommendations; however, low intakes are also observed in special age groups such as adolescents of high-income countries [23,24,25,26].

A review reporting the global mean dietary calcium supplies from FAO balance sheets of different countries shows that in 2011, the global calcium supply at population level was 684 SD± 211 mg per person per day [27]. Using these data, the review also calculated that between 1992 and 2011, the global calcium risk of deficiency decreased from 76% to 51% and that most of those countries at risk (90%) in 2011 were located in Africa and Asia.

A review including studies reporting calcium dietary intake found that the average national dietary calcium intake ranges from 175 to 1233 mg/day (78 studies from 74 countries). Many countries in Asia have an average dietary calcium intake of less than 500 mg/day. Countries in Africa and South America mostly have low calcium intake, between about 400 and 700 mg/day, although there was no information for many countries [28].

Similarly, a recent systematic review of diets in 195 countries based on nationally or subnationally representative nutrition surveys jointly with many dietary data sources of the Global Health Data Exchange for nationally or subnationally representative nutrition surveys provides information about age-standardised intake of dietary factors among adults aged 25 years or older at the global and regional level in 2017. These data show an average global calcium daily intake around 400 mg/day. Lower values were seen in Sub-Sharan African countries and in Southeast Asia, with figures around 200 mg/day. In high-income-countries (HICs), calcium daily intake was around 600–800 mg/day [29].

A review reporting dietary intakes of pregnant women shows consistently low calcium intakes across Asian, African and Latin American countries (105 studies, 73,958 pregnant women from 37 countries) [24,26]. The mean calcium intake in pregnant women of low- and middle-income countries was 648 mg/day (95% confidence interval (CI) 569–727) whereas that from high-income countries was 948 mg/day (95% CI 872–1024) [30,31].

## 5. Calcium and Blood Pressure

Epidemiological studies associated dietary calcium with blood pressure in deprived communities of Guatemala [3]. This population, with poor food intake and limited access to prenatal care, had an incidence of preeclampsia and eclampsia comparable to that of populations with higher resources [32]. Despite having a diet short on nutrients, they had a relatively high intake of calcium. Their diet is based on corn tortillas prepared according to the Mayan tradition that consists of cooking corn with limestone and leave it to soak overnight in hot water [3]. In this way, the grains of corn increase their calcium content, improving their nutritional value. After milling, the flour obtained from grains that underwent this process had an average of 196 mg of calcium per 100 g, while corn flour commonly contains between 10 and 15 mg of calcium per 100 g.

Observational studies also reported an inverse association between water hardness and cardiovascular diseases [33,34,35]. Water hardness is determined by minerals in water, firstly calcium and secondly magnesium [36]. In 1972, the WHO published a general review of these findings [37]. However, later studies showed a weak or inconsistent relationship, possibly because the contribution of calcium from hard water to the total calcium intake was considerably smaller in the populations examined or because more significant risk factors of cardiovascular disease than that of a low dietary calcium intake existed [33].

Studies in animals and humans have shown an inverse relationship between calcium intake and blood pressure [38,39]. Normotensive rats fed a free-calcium diet significantly increased their systolic blood pressure (SBP) between 15 to 35 mmHg in comparison with rats fed a normal calcium diet [38,40,41]. On the other hand, normotensive and hypertensive rats supplemented with calcium had significantly lower values of SBP [42,43,44,45]. A systematic review has shown that calcium supplementation reduces SBP in normotensive adults by 1.14 mmHg (95% CI: −2.01 to −0.27) with doses of calcium of 1000 to 1500 mg/day and by 2.79 mmHg (95% CI: −4.71 to −0.86) with doses of calcium equal to or over 1500 mg/day [46]. In this review, it was found that calcium supplementation had the greatest effect in young adults of less than 35 years, as their systolic blood pressure was reduced by 2.11 mmHg (95% CI: −3.58 to −0.64). A similar systematic review in hypertensive adults found that calcium supplementation reduced SBP by −1.86 mm Hg (95% CI: −2.91 to −0.81) and diastolic BP (DBP) by −0.99 mm Hg (−1.61 to −0.37). However, higher reductions were found in people with a relatively low calcium intake (less or equal to 800 mg/day), in which calcium supplementation reduced SBP by −2.63 (−4.03 to −1.24) and DBP by −1.30 (−2.13 to −0.47) [47]. Another similar systematic review also showed that calcium supplementation as compared to control induced a statistically significant reduction of SBP (mean difference: −2.5 mmHg, 95% CI: −4.5 to −0.6, I^2^ = 42%) but not of DBP (mean difference: −0.8 mmHg, 95% CI: −2.1 to 0.4, I^2^ = 48%) [48].

A systematic review that included 13 RCTs and 15730 pregnant women estimated that calcium supplementation compared to placebo reduced the high blood pressure relative risk (RR) to 0.65, (95% CI: 0.53 to 0.81) and, although with low quality of evidence, also the risk of preeclampsia by 55%, with RR 0.45, (95% CI: 0.31 to 0.65) [49]. In populations with low calcium intake below 800 mg/day, the effect was even higher (RR 0.36, 95% CI: 0.20 to 0.65). A recent RCT has shown a reduction in the incidence of preeclampsia of 34% (RR 0.66, 95% CI: 0.44–0.98) in women supplemented with 500 mg/day of calcium before and in early pregnancy, comparing groups of women with good adherence to supplement intake, either calcium or placebo [50]. Both groups received calcium supplementation corresponding to 1.5 g/day after 20 weeks´ gestation.

## 6. Calcium Intake and Effect on Blood Pressure

Blood pressure is regulated by intracellular calcium in vascular smooth muscle cells, through vasoconstriction and variations of the vascular volume [51,52]. Low calcium intake seems to trigger both mechanisms, raising plasma parathyroid hormone (PTH) levels, that increase intracellular calcium directly or through calcitriol activation, and stimulating the renin–angiotensin–aldosterone signalling pathway that produces sodium and water reabsorption, thus increasing the vascular volume [42,52,53,54,55]. Parathyroidectomized rats in comparison with sham-operated rats did not show an increase in blood pressure after 10 weeks of a calcium-free diet [54].

There is no established threshold for the benefits of calcium intake on blood pressure; in humans and animals with low calcium intake, blood pressure is improved when calcium intake is increased to reach the recommended levels. No benefits for blood pressure by increased calcium intake have been observed in humans and animals with adequate calcium intake [46,49,56,57]

## 7. Calcium Intake during Pregnancy and the Effects on the Offspring

The effect of calcium supplementation during pregnancy has also been explored. The follow-up of children whose mothers were involved in a RCT of calcium supplementation showed that children whose mothers were in the calcium group had a reduction in the risk of high blood pressure (above the 90th percentile) at seven years of age in comparison with children whose mothers were in the placebo group (RR 0.59; 95% CI: 0.39 to 0.90) [56]. A systematic review shows that children whose mothers received calcium supplementation had a reduction of −1.92 mm Hg (95% CI −3.14 to −0.71) in SBP at age 1 to 9 years [58].

A follow-up until 52 weeks of age of offspring of rats whose mothers had a low calcium intake during pregnancy showed values of SBP of 12.1 mmHg (95% CI: 8.8 to 15.4, *p* < 0.0001) higher than the offspring of rats whose mothers had a normal calcium diet during pregnancy [59].

In the RCT mentioned above, children whose mothers were in the calcium group showed a 27% reduction in the risk of developing dental caries at 12 years of age in comparison with children whose mothers were in the placebo group (RR: 0.73, CI 95%: 0.62; 0.87) [60].

## 8. Other Effects of Calcium Intake on Health

An adequate dietary calcium intake has been associated not only with the prevention of hypertensive disorders of pregnancy and blood pressure reduction but also with low-density lipoprotein (LDL) cholesterol levels and prevention of osteoporosis and colorectal adenomas [13,61,62].

### 8.1. Cholesterol

A systematic review of calcium supplementation and lipid metabolism reported that calcium supplementation reduced LDL cholesterol [−0.12 mmol/L (95% CI: −0.22 to −0.02)] and increased high-density lipoproteins (HDL) cholesterol [0.05 mmol/L (95% CI: 0.00 to 0.10) [63]. The authors explained that possible mechanisms of these effects by the increase in dietary calcium include the suppression of calcitrophic hormones that reduce intracellular calcium in adipocytes, thus stimulating lipogenesis and lipid storage [63]. Besides, dietary calcium may decrease serum cholesterol by inhibiting cholesterol and saturated fatty acid absorption [64].

### 8.2. Bone Health

A systematic review from 2006 that included 19 studies involving 2859 children found that calcium supplementation had a small effect on total body bone mineral content (standardised mean difference 0.14, 95% CI: 0.01 to 0.27) and upper limb bone mineral density (0.14, 0.04 to 0.24), and this effect persisted after the end of supplementation only for the upper limb bone mineral density (0.14, 0.01 to 0.28) [65]. The benefits of calcium supplementation seem to be greater in children and adolescents with low calcium intake [66]. Calcium effects in other age groups were usually evaluated in combination with vitamin D, so data for calcium alone are limited [17].

The US preventive Task Force in 2013 did not recommend supplementation with vitamin D or calcium for the prevention of fractures in community-dwelling adults; however, the evidence was updated in 2016, and with newer data from the Women´s Health Initiative study, the new meta-analysis showed a 15% reduction on the incidence of fractures and a 30% reduction in hip fractures in middle-aged to older adults [67,68].

### 8.3. Recurrent Colorectal Adenomas

A systematic review of randomised controlled trials found that calcium supplementation with doses from 1200 to 2000 mg/day and treatment duration from 36 to 60 months reduced the risk of recurrent colorectal adenomas (RR = 0.89, 95% CI: 0.82–0.96, 5 studies, 2984 participants) [69].

It was proposed that calcium binds bile acids in the bowel lumen, inhibiting their proliferative and carcinogenic effects [70]. In support of this hypothesis, studies in animals have indicated a protective effect of dietary calcium on bile-induced mucosal damage and experimental bowel carcinogenesis [71,72].

## 9. Calcium Supplementation Concerns

### 9.1. Supplements and Renal Stones

There is some controversy as to whether increasing calcium intake increases the risk of kidney stone formation, as one RCT that evaluated the effect of 500 mg of calcium plus 200 units of Vitamin D3 on the risk of fracture in postmenopausal women showed an increased risk of renal stones (hazard ratio, 1.17; 95% CI: 1.02 to 1.34) [73]. We selected studies included in a systematic review of calcium supplementation and incidence of kidney stones that evaluated an intervention consisting of calcium supplementation alone in non-pregnant subjects [74]. We found five studies in postmenopausal or elderly women with a total of 2038 subjects, randomised to calcium or placebo. We meta-analysed the results and found a null effect of calcium supplementation compared to placebo (RR 0.66, 95% CI: 0.19, 2.34). 

A systematic review showed that calcium supplementation during pregnancy did not increase the risk of urolithiasis (RR 1.52, 95% CI: 0.06, 40.67) or renal colic (RR 1.75, 95% CI: 0.51, 5.99) as shown in two studies with 12901 women [75].

Moreover, a secondary analysis of 7982 women with a history of nephrolithiasis participating in a prospective study found that the proportion of calcium absorption decreased with the increase of both dietary and supplement calcium intake and that increased calcium intake reduced the likelihood of nephrolithiasis by 45–54% (*p =* 0.03) [76].

Nowadays, dietary calcium restriction is not recommended for stone formers with nephrolithiasis; on the contrary, diets with more than 1 gram of calcium per day could be protective against stone formation [77]. Kidney stones are mainly composed of calcium combined with oxalate or phosphate [78]. The calcium remaining in the intestine would impede the absorption of products associated with the risk of renal stones, such as oxalates, and the intake of calcium supplements during meals would decrease the absorption of oxalates and thus the formation of stones [74].

### 9.2. Calcium Supplements and Myocardial Infarction

Calcium supplements are commonly used to prevent fracture in postmenopausal women [1,7]. However, the use of calcium supplements in postmenopausal women was discouraged after a narrative review showing a potential increase of adverse events such as atherosclerotic vascular disease in women from New Zealand receiving calcium supplements [79].

However, a newer systematic review and meta-analysis including 18 RCTs involving 63,564 elderly women participants concluded that calcium supplementation with or without vitamin D had no effect on coronary heart disease or all-cause mortality risk [80,81,82]. This review reported cardiovascular events clinically verified by hospital records or death certificates and found that the RR for all-cause mortality was 0.96 (95% CI: 0.91–1.02; *p* = 0.180), that for CHD events was 1.02 (95% CI: 0.96–1.09; *p* = 0.510) and that for myocardial infarction was 1.08 (95% CI: 0.92–1.26; *p* = 0.320).

### 9.3. Calcium Supplements and Iron Absorption

There have been concerns related to the effects of calcium supplements on iron absorption based on short-term studies reporting that calcium supplements inhibit iron absorption by 28 to 55% depending on the dose, type of salt used, time of supplementation and the presence in the food of hem or non-hem iron [83]. However, evidence shows no effect on iron status of prolonged calcium supplementation taken at the same time or separate of meals [84,85,86,87,88,89]. A study of infants supplemented with calcium glycerophosphate or placebo found no difference in iron status at four and nine months, with mean changes in serum ferritin of −24.5 and −46.6 μg/L in intervention and placebo groups, respectively [90]. Another study in which adolescent girls received 1 g of calcium citrate malate/day for 4 years did not find differences in the iron status; serum ferritin average concentrations in the supplemented group at baseline and years 1, 2, 3 and 4 were 29.1 ± 1.3, 31.1 ± 1.5, 31.1 ± 1.6, 30.6 ± 2.0 and 29.6 ± 1.9 μg/L, respectively, whereas in the placebo group, the concentrations were 29.3 ± 1.4, 33.8 ± 1.7, 32.3 ± 1.4, 30.9 ± 1.5 and 29.5 ± 1.6 μg/L, respectively (*p* = 0.88, 0.23, 0.56, 0.88 and 0.96 for baseline and years 1, 2, 3 and 4, respectively) [91]. Similarly, another study of 113 adolescent girls supplemented with 500 mg of calcium/day as calcium carbonate found no differences in iron status markers. At one year, hemoglobin was 136 g/L in the supplemented group and 134 g/L in the placebo group (*p =* 0.31); ferritin was 25.4 μg/L in the supplemented group and 26.1 μg/L in the placebo group (*p* = 0.73) [88].

A study that supplemented post-partum women with 500 mg of calcium/day as calcium carbonate found that mean serum ferritin was 28.4 mg/L in the supplemented participants and 27.5 mg/L in women in the placebo group (*p* > 0.5) [87]. A study in 24 healthy individuals also showed that supplementation with 1200 mg of calcium/day as calcium carbonate had no significant effect on hemoglobin or hematocrit at 6 months. Mean hemoglobin at 6 months was 136 g/L in the calcium group compared to 139 ± 4 in the control group and hematocrit was 0.416 ± 0.013 in the calcium group compared to 0.424 ± 0.009 in the control group [92].

With this evidence, it has also been recommended that pregnant women take calcium supplementation together with iron and folic acid to improve adherence [93].

### 9.4. Calcium Intake and Maternal Bone Post-Partum Resorption

Of concern are the findings in a RCT in rural Gambian women with very low calcium intake showing that supplementation during pregnancy with 1.5 g calcium/day resulted in significantly lower maternal bone mineral content, bone area, and bone mineral density at the hip throughout 12 months of lactation compared to women in the placebo group [94]. These women also experienced greater decreases in bone minerals during lactation at the lumbar spine and distal radius and had biochemical changes consistent with greater bone mineral mobilization that could last for a long time [94,95]. The authors postulated that, possibly, calcium supplementation disrupted the processes of calcium conservation previously seen in these women with a very low calcium diet and that the withdrawal of a calcium supplement produced a rise in parathyroid hormone (PTH) secretion, promoting renal calcium reabsorption, intestinal calcium absorption and bone resorption [94]. However, RCTs of calcium supplementation during pregnancy performed in other populations with low and normal calcium intake in the USA, Mexico and Brazil showed contradictory results, since women in the calcium group showed reduced postpartum bone resorption and improved bone recovery [96,97,98,99].

## 10. Calcium Intake and Drug Interactions

There are some reports showing interactions of calcium supplements with drugs, with the majority of them having a moderate or minor Interaction Rating. Major interactions have been reported with antiretroviral drugs like Dolutegravir and Elvitegravir, and the recommendation is to take these drugs separately from calcium supplements. Calcium can decrease the absorption of some antibiotics, and the recommendation is to take calcium supplements apart from these antibiotics. Besides, supplements with high amounts of calcium are not advised when taking some calcium channel blockers drugs [100]. As we will further develop in this manuscript, our recommendation is to increase calcium intake through diet by eating calcium-rich foods or calcium-fortified foods, as calcium supplementation does not look like a feasible strategy to increase calcium intake in all populations.

## 11. Discussion

From the above information, it can be concluded that adequate calcium intake has many health benefits besides its favourable effects on bone health, and action should be taken to ensure an adequate calcium intake (Table 2). The effect of calcium supplementation on pregnancy outcomes is the one with more evidence as it received more research attention. However, the effect of calcium supplementation on lowering blood pressure, particularly at an early age, is also very important for its impact on the prevention of cardiovascular complications later in life. Further studies are required to better understand the modeling effect of calcium intake during pregnancy on progeny blood pressure.

Other benefits of adequate calcium intake have been reported, such as higher bone mineral accretion at early ages and prevention of osteoporosis and colorectal cancer.

Many of the reported deleterious effects of calcium supplementation were recently reviewed, and previous results were questioned. A recent meta-analysis showed no effect on coronary heart disease or all-cause mortality risk in postmenopausal women supplemented with calcium. Long-term calcium supplementation did not show a detriment to iron status. The effect of calcium intake on renal stones formation was shown to be contradictory, and new evidence shows that a diet with an adequate calcium intake actually prevents the formation of calcium stones.

Of concern are the calcium resorption outcomes reported in the follow-up of Gambian women that had received calcium supplementation during pregnancy. Whereas other trials have shown opposite results to those of the Gambian study, more research on the effects of calcium supplementation on bone health after lactation is required to better understand these particular findings.

All this evidence suggests that every subject should attain an adequate calcium intake. Inequities in calcium intake are striking. Strategies to improve calcium intakes should be evaluated according to the target population.

Individuals with low calcium intake should be counselled on the importance of calcium intake and guided on achieving an adequate intake, especially if belonging to high-risk groups such as children, adolescents and women (with emphasis on the reproductive period). Populations at risk of low calcium intake should be identified, and strategies should be designed according to each particular situation.

Efforts should be made to achieve calcium recommendations both at individual and global levels, and strategies will depend on the intake level of each population. Calcium intakes in low- and middle-income countries (LMICs) are extremely inadequate, and strategies should include the whole population; however, even in high income countries, certain population groups such as pregnant women often do not meet the recommendations, and strategies should probably be targeted specifically to those groups.

Supplementing with calcium individuals that already reach the requirements has been shown to be of no benefit. Even in the NICHD/NIH study that showed no overall effect of calcium supplementation in women with high basal intake (mean 1114 mg/day), there was a tendency towards a lower incidence of preeclampsia according to the quintiles of basal calcium intake [101]. A decrease in preeclampsia incidence, although not statistically significant, was observed in women in the quintiles of calcium intake below the requirements [101].

Another point to discuss regards the amount of calcium these strategies should provide to achieve an adequate intake. Theoretically, this amount should be determined after assessment of each specific population. A review of calcium intake during pregnancy showed an average difference in calcium intake of around 400–500 mg/day between LMICs and HICs. WHO guidelines recommend calcium supplementation of 1.5 to 2 g/day to pregnant women from populations with low basal intake; however, a newer review showed that supplementation with 500 mg/day of calcium during pregnancy had effects similar to those of supplementation with higher doses [102].

There are three broad approaches to improve dietary calcium intake: one is a behavioural intervention that, although ideal, relies on personal habits and abilities, the second one is supplementation that targets individuals, and the third one is food fortification that aims at improving the dietary intake of a whole population. Recommendations to improve dietary calcium intake by increasing the consumption of calcium-rich foods and/or taking calcium supplements have been around for many years; however, these recommendations have shown little impact in LMICs.

Supplementation strategies, though well-evaluated in research studies, have many limitations for implementation in LMICs, such as end user costs, poor access to the health system and low long-term compliance. In addition, some deleterious effects discussed above, such as post-partum bone resorption, gastrointestinal discomfort and cardiovascular effects on adult women, as well as some drug interactions have been reported in subjects taking calcium supplements.

Food fortification looks like as a promising approach, since it can reach different age groups for a long period of time and, if properly designed, does not require changing the dietary habits or taking pills, which allows to reach populations outside the healthcare system.

The calcium dietary intake gap between LMICs and HICs is around 400–500 mg/day, a feasible amount to achieve with food fortification. Taking into account the benefits shown during pregnancy on mother and foetus and the benefits on bone mineral accretion and prevention of bone loss, it can be assumed that achieving an adequate calcium intake at population level would imply long-lasting benefits and will reduce the differences in many health outcomes between LMICs and HICs.

Food fortification has been used for more than 80 years [103]. Mandatory food fortification has contributed to health improvement, lowering the incidence of goitre, beriberi and pellagra. Currently, more than 130 countries have mandatory micronutrient fortification of salt, and around 85 have mandatory micronutrient fortification of wheat flour. WHO evaluation of food fortification nutrition interventions recommends the fortification of maize flour with iron and folic acid and the fortification of salt with iodine and iron powders [104]. Improved zinc, vitamin A, folic acid, vitamin D and calcium deficiencies at population level were also achieved by fortifying different staple foods. Besides fortification, the restoration of micronutrients naturally present in foods that are removed during industrialization processes, such as vitamin B complex in maize flour, is used. The restoration of calcium to wheat flour has been used in the UK since 1943. However, despite this vast experience, food fortification products are not always accessible to the low socioeconomic groups, and strategies should be developed to reach them [103].

Most experiences include staple food fortification, as they are generally consumed in good amounts, and research is required to determine the best staple food to be fortified for each specific population. On the other hand, water fortification looks promising, since water is universally consumed and taking into account the global obesity figures water does not imply any change in energy intake.

Ecological studies have found an inverse relationship between water hardness and cardiovascular mortality [33,105]. Calcium bioavailability from calcium-rich waters is similar to that of milk [13,106,107]. Water fortification with calcium would in some way mimic the situation of populations drinking hard water. Although there are some studies on this topic, more research is required. Studies on water fortification include examples of fortification of public water supplies with fluoride to prevent dental cavities that have been implemented for more than 50 years in more than 25 countries; however, most evidence of its effects comes from observational studies [108,109,110,111]. There are some studies in Asia regarding iodized water; however, as iodine has limited stability, this strategy is not always cost-effective [103]. Iron and ascorbic acid water fortification to prevent iron deficiency anaemia has been explored and, although successful, is still being researched [112,113].

It has been estimated that in LMICs, improved drinking water can reach the majority of urban areas (92%), and 70% of the urban population has access to piped water within their household. In contrast, in rural areas, only 25% of the population has access to this type of service [114]. Research into the feasibility of calcium-fortified water deserves to be considered, taking into account different water supplies.

## 12. Conclusions

Achieving calcium intake recommendations could involve major health benefits to individuals and populations and that it is fully justified to make efforts to attain such achievement. As in the majority of health situations, inequities in calcium intake are clearly shown at the global level. Regarding the achievement of calcium recommendations at the population level, long-term beneficial effects can be expected, including the improvement of health in future generations.

## Figures and Tables

**Table 1 nutrients-11-01606-t001:** Dietary reference values for Calcium from different sources.

	UK (SACN) [15]		USA and Canada (IOM) [15]		FAO/WHO [13]		European (EFSA) [14]	
Age	Estimated Average Requirement (mg/day)	Recommended Nutrient Intake	Estimated Average Requirement (mg/day)	Recommended Dietary Allowance (mg/day)	Estimated Average Requirement (mg/day)	Recommended Nutrient Intake	Average Requirement (mg/day)	Population Reference Intake
0–6 month	400	525		200 (AI)	240–300	300–400		
6–12 month	400	525		260 (AI)	240–300	300–400		280 (AI)
1–3 year	275	350	500	700		500	390	450
4–6 year	350	450	800	1000	440	600	680	800
7–10 year	425	550	800	1000		1300	680	800
Males								
11–14 year	750	1000	1100	1300	1040	1300	960	1150
15–18 year	750	1000	1100	1300	1040	1300	960	1150
19–24 year	525	700	800	1000	840	1000	860	1000
25–50 year	525	700	800	1000	840	1000	750	950
50 year	525	700	800	1000	840	1000/1300	750	950
Females								
11–14 year	625	800	1100	1300	1040	1300	960	1150
15–18 year	625	800	1100	1300	1040	1300	960	1150
19–24 year	525	700	800	1000	840	1000	860	1000
25–50 year	525	700	800	1000	840	1000	750	950
50 year	525	700	1000	1200	840	1000	750	950
Pregnancy								
14 to 18 year	Same as non-pregnant	Same as non-pregnant	1100	1300	*	*	Same as non-pregnant	Same as non-pregnant
19 and older	Same as non-pregnant	Same as non-pregnant	800	1000	940	1200	Same as non-pregnant	Same as non-pregnant
Lactation	plus 550	plus 550	1100/800	1300/1000	1040	1000	Same as non-lactating	Same as non-lactating

SACN: UK Scientific Advisory Committee on Nutrition. IOM: USA Institute of Medicine. EFSA: European Food Safety Authority; AI: Average Intake. * No data available.

**Table 2 nutrients-11-01606-t002:** Effect of calcium intake on health outcomes. Evidence from randomised controlled trials (RCT) and systematic reviews of randomised controlled trials.

Health Outcomes	Outcome	Population Group	Research Evidence	Effect Size
Hypertensive disorders of pregnancy	Preeclampsia	Pregnant women	Meta-Analysis	Calcium supplementation compared to placebo reduced the risk of preeclampsia, RR 0.45, (95% CI: 0.31 to 0.65) [48].
		Pregnant women with low basal calcium intake	Meta-Analysis	Calcium supplementation compared to placebo reduced the risk of preeclampsia, RR 0.36, (95% CI: 0.20 to 0.65) [48].
	High blood pressure	Pregnant women	Meta-Analysis	Calcium supplementation compared to placebo reduced the high blood pressure relative risk (RR) to 0.65, (95% CI: 0.53 to 0.81) [48].
Blood pressure	Blood pressure	Normotensive adults	Meta-Analysis	Calcium supplementation reduced systolic blood pressure (SBP) in adults by 1.14 mmHg (95% CI: −2.01 to −0.27) with doses of calcium 1000 to 1500 mg/day and by 2.79 mmHg (95% CI: −4.71 to −0.86) with doses of calcium equal to or over 1500 mg/day. Calcium supplementation had the greatest effect in young adults of less than 35 years as their SBP was reduced by 2.11 mmHg (95%CI: −3.58 to −0.64) [45].
	Blood pressure	Hypertensive adults		Calcium supplementation reduced SBP by −1.86 mm Hg (95% CI: −2.91 to −0.81) and diastolic BP (DBP) by −0.99 mm Hg (95% CI: −1.61 to −0.37) [46].
	Blood pressure	Hypertensive adults with low basal calcium intake		In people with relatively low calcium intake (≤ 800 mg per day) calcium supplementation reduced SBP by −2.63 (95% CI: −4.03 to −1.24) and DBP by −1.30 (95% CI: −2.13 to −0.47) [46].
	Blood pressure	Hypertensive adults		Calcium supplementation as compared to control induced a statistically significant reduction in SBP (mean difference: −2.5 mmHg, 95% CI: −4.5 to −0.6, I(2)= 42%) but not DBP (mean difference: −0.8 mmHg, 95% CI: −2.1 to 0.4, I(2) = 48%) [47].
Progeny blood pressure	High blood pressure	Pregnant women/children	RCT	Calcium supplementation showed that children whose mothers received calcium supplementation had, at seven years of age, a reduction in the risk of high blood pressure (above the 90th percentile) in comparison with children whose mothers were in the placebo group (RR 0.59; 95% CI: 0.39 to 0.90) [55].
Cholesterol	LDL and HDL Cholesterol	Adults	Meta-Analysis	Calcium supplementation reduced low-density lipoprotein (LDL) cholesterol [−0.12 mmol/L (95% CI: −0.22 to −0.02)] and increased high-density lipoprotein (HDL) cholesterol [0.05 mmol/L (95% CI: 0.00 to 0.10) [59].
Colorectal adenomas	Recurrent colorectal adenomas	Adults with previous adenomas	Meta-Analysis	Calcium supplementation with doses from 1200 to 2000 mg/day and treatment duration from 36 to 60 months reduced the risk of recurrent colorectal adenomas, RR = 0.89, (95%CI: 0.82–0.96) [65].
Bone health	Bone mineral density	Children	Meta-Analysis	Calcium supplementation had a small effect on total body bone mineral content (standardised mean difference 0.14, 95% CI: 0.01 to 0.27) and upper limb bone mineral density (0.14, 95% CI: 0.04 to 0.24), and this effect persisted after the end of supplementation only in the upper limb (0.14, 95% CI: 0.01 to 0.28) [61].
Renal stones	Urolithiasis	Individuals with osteoporosis	Meta-Analysis	Calcium supplementation compared to placebo, RR 0.66 [95% CI 0.19, 2.34]; 5 studies in postmenopausal or elderly women including 2038 subjects [70].
	Urolithiasis	Pregnant women	Meta-Analysis	Calcium supplementation during pregnancy did not increase the risk of urolithiasis, RR 1.52 [95% CI: 0.06, 40.67] or renal colic, RR 1.75 [95% CI; 0.51, 5.99] in 2 studies with 12901 women [71].

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
