# Peer review of "Calcium Intake and Health"

_nutrients, 2019, doi:10.3390/nu11071606_

Reviewer 1 Report

June 13, 2019

Comments and suggestions for authors

In this manuscript, Cormick et al. summarize the importance of calcium intake for human body, and the benefits from appropriate calcium intake. This short review will benefit to community health and could help people to be away from calcium deficiency inducing disorders.

I have two important recommendations and one minor typo:

Recommendations:

1.     Please consider to add a list of calcium content of foods which can help reader to easily be guided in daily dietary.

2.     Please add one more section of “Calcium intake and drug interactions”. This is very important.

Minor typo:

3.     On page 1, at the end of the first paragraph, it should be 400-500 mg/day.

Reviewer 2 Report

This review represents a thorough, updated compilation of the role of calcium in human health.  Table 2 is especially useful.  Meta-analyses were used when available to support relationships.

1.  p. 4 2nd line  Not ‘maintain health’ but ‘health maintenance’.

2.  p. 4 Global Calcium Intake.  Inequities.

     2nd paragraph  Not ‘Using this data’ but ‘Using these data’.

3.  p. 5 4th paragraph  Describe calcium as a threshold nutrient to explain why the benefit occurs

     in those with low baseline intakes.

4.  p. 10 6th paragraph  ‘the discussion is on reaching the recommendations in people with low

     calcium intake or supplementing with calcium without taking into account the basal intake’. 

     Needs rewriting. 

     What discussion?  Who is not taking into account basal intake?  Why is it important?
